# LeVERB: Humanoid Whole-Body Control with Latent Vision-Language Instruction

## Abstract

Vision–language–action (VLA) models have demonstrated strong semantic understanding and zero-shot generalization, yet most existing systems assume an accurate low-level controller with hand-crafted action "vocabulary" such as end-effector pose or root velocity. This assumption confines prior work to quasi-static tasks and precludes the agile, whole-body behaviors required by humanoid whole-body control (WBC) tasks. To capture this gap in the literature, we start by introducing the first sim-to-real-ready, vision-language, closed-loop benchmark for humanoid WBC, comprising over 150 tasks from 10 categories. We then propose LeVERB: Latent Vision-Language-Encoded Robot Behavior, a hierarchical latent instruction-following framework for humanoid vision-language WBC, the first of its kind. At the top level, a vision–language policy learns a latent action vocabulary from synthetically rendered kinematic demonstrations; at the low level, a reinforcement-learned WBC policy consumes these latent verbs to generate dynamics-level commands. In our benchmark, LeVERB can zero-shot attain a 80% success rate on simple visual navigation tasks, and 58.5% success rate overall, outperforming naive hierarchical whole-body VLA implementation by 7.8 times.

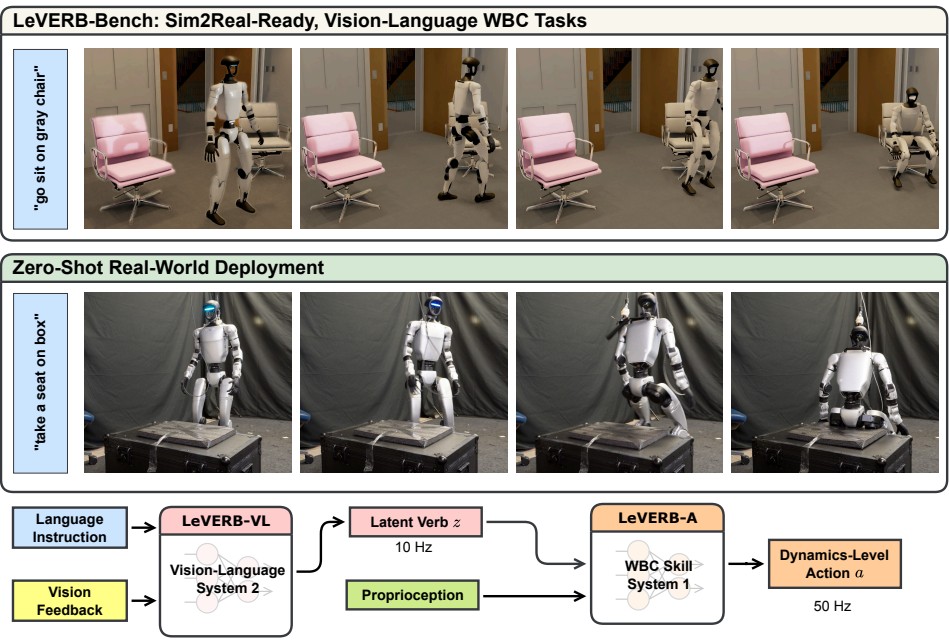

Figure 1: Overview of our contributions. Top: we create a photorealistic and dynamically accurate benchmark for humanoid vision-language WBC. Middle: in real world, we zero-shot deploy a dual-process VLA model trained only on synthetic data. Bottom: a high-level overview of our model architecture with decoupled vision-language and dynamics-level action processing.

# 1 INTRODUCTION

Enabling humanoid robots to perceive complex scenes, interpret intuitive language commands, and execute whole-body actions has long been a captivating yet technically challenging goal. Recent advances in Vision-Language-Action (VLA) models have demonstrated promising capabilities in complex tabletop and mobile manipulation (Brohan et al., 2023; Kim et al.; Team et al., 2025; Black et al.), and embodied navigation (Cheng et al., 2024a; Shah et al., 2023; Xu et al.) tasks, leveraging their semantic reasoning capacity to bridge perception, language, and control. However, applying VLAs to humanoid robots remains underexplored. In contrast to quasi-static arm-based manipulators, humanoid robots are inherently high-dimensional nonlinear dynamic systems.

Humans think both fast and slow. Vision and language, processed ultimately in the cortex, enable high-level reasoning and long-horizon planning, while sensory-motor responses, mediated by spinal reflexes and subcortical motor circuits, supports rapid, reactive control (Jensen, 2006). This dual-process architecture allows humans to execute complex motor skills while adapting to changing environments. To achieve comparable whole-body competence, humanoid robots - which share similar complexity in motor control - must likewise integrate a hierarchical architecture combining high-frequency control from proprioceptive feedback with low-frequency planning and semantic reasoning grounded in rich vision and language inputs.

Prior works on VLA-enabled WBC rely on explicit, low-dimensional action "vocabulary", such as base velocities, end-effector pose, etc., as the interface between VLA models and low-level controllers (Cheng et al., 2024a; Ding et al., 2025), where the low-level control is only capable of a few atomic skills and is designed to rapidly react to fine-grained high-level commands. However, such interfaces restrict expressiveness and make it difficult to integrate complex whole-body motions and scene interactions. To fully unleash humanoid whole-body capabilities, we need (1) a learned "latent vocabulary" that is expressive enough to both cover whole-body motions and capture semantics encoded in the vision and language inputs, and couple it with (2) a versatile WBC layer that dynamically translates this vocabulary into humanoid-feasible actions that are zero-shot transferrable to the real world.

To bridge this gap, we introduce LeVERB, Latent Vision-Language Encoded Robotic Behavior, the first vision-language latent action model for humanoid whole-body control. LeVERB consists of a high-level vision-language policy (*System 2*) that interprets vision-language inputs, and a low-level reactive controller (*System 1*) to execute whole-body motions. To overcome the scarcity of robot-specific visual data, we develop a data synthesis pipeline that collects diverse human motions retargeted to the humanoid robot and renders them photorealistically in randomized scene contexts. A set of semantically similar language commands are then annotated using a VLM. This enables training the high-level VLA directly on paired, robot-specific video and language data.

To learn a structured latent space from vision-language inputs, we propose a CVAE-based architecture for the high-level VLA module. This structured latent space is key to learning a unified vision-language-action distribution that accurately aligns perception and action while mitigating overfitting. To reduce the cost of photorealistic rendering during parallel simulation training, we decouple the learning process. We first train the vision-language component using kinematics reconstruction to align visual and motion semantics. Then, we freeze its latent space and train a separate action module that samples from it to learn a proprioception-only controller focused on mastering robot dynamics.

Trained entirely on synthetic data, LeVERB enables flexible, instruction-driven humanoid behavior. It can follow commands grounded in both state-space objectives (e.g., "turn left", "walk straight") and visual goals (e.g., "go to the table in front", "sit on the green chair"). At inference, the vision-language module (*System 2*) encodes vision and language inputs into a latent action plan, which is then decoded by the low-level controller (*System 1*) into motor commands executable on the robot. We demonstrate that LeVERB achieves zero-shot closed-loop deployment, executing expressive whole-body motions and scene interactions in simulation and on real humanoid hardware.

Our work advances the field of VLA-driven WBC in three significant ways. First, we develop a scalable, ready-to-use synthetic data generation pipeline that renders robot kinematic motions with photorealistic rendering and diverse scene randomization for training vision-language models for humanoid robots, including a closed-loop dynamic environment for evaluation. Further, we propose a novel CVAE-based hierarchical vision-language policy that learns a structured latent space, enabling

semantically grounded whole-body behaviors from high-level visual and language inputs. Finally, we validate our approach both in simulation and on real-world humanoid hardware, demonstrating generalization to unseen scenarios and establishing the first zero-shot sim-to-real results for WBC using a latent vision-language interface.

## 2 RELATED WORK

### 2.1 HUMANOID WHOLE BODY CONTROL

Recent advances in physics-based animation have shown strong results in humanoid whole-body control, primarily through motion tracking (Peng et al., 2018; 2021; Luo et al., 2023a), where reinforcement learning (RL) policies imitate reference motions from human MoCap data. Building on this, methods like PULSE (Luo et al., 2023b) and MaskedMimic (Tessler et al., 2024) learn latent policies controllable by high-level inputs, using Conditional VAEs or Transformers conditioned on language and object interactions. TokenHSI (Pan et al., 2025) further supports compositional object interactions. However, these methods depend on privileged simulation states (e.g., full object poses), limiting real-world applicability and ignoring visual inputs.

In contrast, real-world humanoid systems avoid latent-conditioned low-level control, instead predicting explicit commands such as base velocity, orientation, or whole-body keyframes (Cheng et al., 2024b; Li et al., 2025a; He et al., 2024; Fu et al., 2024; Ji et al., 2024). This modularity eases integration but often results in jittery or unnatural motions due to infeasible predictions. LangWBC (Shao et al., 2025) introduces a language-conditioned CVAE for whole-body control with latent structure but lacks visual grounding and high-level reasoning, limiting it to simple commands. Incorporating vision-conditioned latent policies remains a key challenge for humanoid control.

### 2.2 HIERARCHICAL VLA FOR ROBOT LEARNING

Recent progress in manipulation (Intelligence et al., 2025; Black et al.; Ye et al., 2024; Zhen et al., 2024; Kim et al.; Team et al., 2024) shows that vision-language-action (VLA) models enable generalization to open-world tasks by integrating visual and linguistic inputs with low-level control. However, end-to-end models often incur high inference latency due to the size of VLA backbones, resulting in delayed or discontinuous motions. This is particularly problematic for humanoid whole-body control, which demands high-frequency, low-latency feedback for stability and agility. To address this, recent works (Bu et al., 2025; Zhang et al.; Bjorck et al., 2025; Han et al., 2024; Li et al., 2025b) adopt hierarchical System-1-System-2 architectures. Notably, AGIbot (Bu et al., 2025) demonstrates that using a latent interface improves performance. For dynamic systems like legged or humanoid robots, prior real-world approaches often rely on explicit interfaces between high-level and low-level policies. For example, Liu et al. (2024a) uses end-effector poses and base velocities for whole-body manipulation; NaVILA (Cheng et al., 2024a) predicts direction and distance for velocity control; and Humanoid-VLA (Ding et al., 2025) forecasts full-body poses, offering expressiveness but requiring task-specific tuning. These explicit strategies simplify modular training but limit generalization to diverse whole-body skills, such as seated interactions.

To our knowledge, no prior work has demonstrated vision-language-driven whole-body control on real humanoid robots using a hierarchical latent architecture—an important gap this work aims to fill.

### 2.3 HUMANOID BENCHMARK

Demonstration data is a critical enabler for training VLA models, but collecting such data for robotic control is nontrivial. In the manipulation domain, recent works have tackled this problem through large-scale data collection pipelines using teleoperation (Kim et al., 2025; Octo Model Team et al., 2024; Black et al.) and expert policy distillation (Niu et al., 2025). In contrast, demonstration data for visual WBC in humanoid robots remains scarce. Although recent efforts have advanced teleoperation for humanoids (Ji et al., 2024; He et al., 2024; Ze et al., 2025), large-scale visual demonstrations have yet to be provided due to the complexity of collecting whole-body motions on physical robots. Existing benchmarks either focus solely on locomotion (Al-Hafez et al., 2023), operate purely in the state space without visuals (Sferrazza et al., 2024; Luo et al., 2024), or have non-photorealistic renderings (Liu et al., 2024b) leading to large sim-to-real gaps. We present the first benchmark and

Figure 2: Visualization of LeVERB-Bench environments. Top row: hundreds of texture and object randomization options. Middle row: egocentric camera view and randomized third-person camera views. Bottom row: diverse task categories.

dataset that provides both photorealistic renderings and physics-based simulation for whole-body motions that are readily transferred to real-world hardware.

## 3 LeVERB Dataset and Benchmark

Since there exists no suitable dataset and benchmark for vision-language-based humanoid WBC, we will first introduce LeVERB-Bench, an efficient and scalable pipeline for synthetic visual-language humanoid WBC data generation and closed-loop benchmarking. An overview of the dataset visualizations is shown in Figure 2.

The main innovation of our efficient synthetic data generation pipeline is to replay retargeted MoCap motions in simulation to collect photorealistic rollouts. This offers three key advantages: (1) it removes the need for reliable dynamic control during data collection, (2) kinematic poses provide sufficient task-level semantics for vision-language understanding, and (3) it supports future use of retargeted humanoid data from sources like internet videos (Goel et al., 2023). As we show later, despite minor artifacts, using kinematic-only rendering is sufficient when paired with a high-quality low-level policy for closed-loop control.

Table 1: Distribution of task categories

| Vision-Language Tasks | | | |
|---|---|---|---|
| **Category** | **# Motions** | **Total [s]** | **Avg [s]** |
| Navigation | 101 | 465.6 | 4.61 |
| Towards | 80 | 372.0 | 4.65 |
| Around | 21 | 93.6 | 4.46 |
| Locomotion | 20 | 64.4 | 3.22 |
| Sitting | 23 | 74.4 | 3.23 |
| Reaching | 10 | 17.4 | 1.74 |
| **Total** | **154** | **621.7** | **4.04** |
| **Language-Only Tasks** | | | |
| **Category** | **# Motions** | **Total [s]** | **Avg [s]** |
| Locomotion | 399 | 1052.8 | 2.6 |
| Reaching | 61 | 101.6 | 1.7 |
| **Total** | **460** | **1154.5** | **2.5** |

We use the ray-tracing rendering in IsaacSim to render our data. This allows more accurate simulation of scene lighting and shadows, alleviating the sim-to-real gap caused by unrealistic lighting in prior works on synthetic data (Bonetto et al., 2023). To create a diverse set of visual scenarios

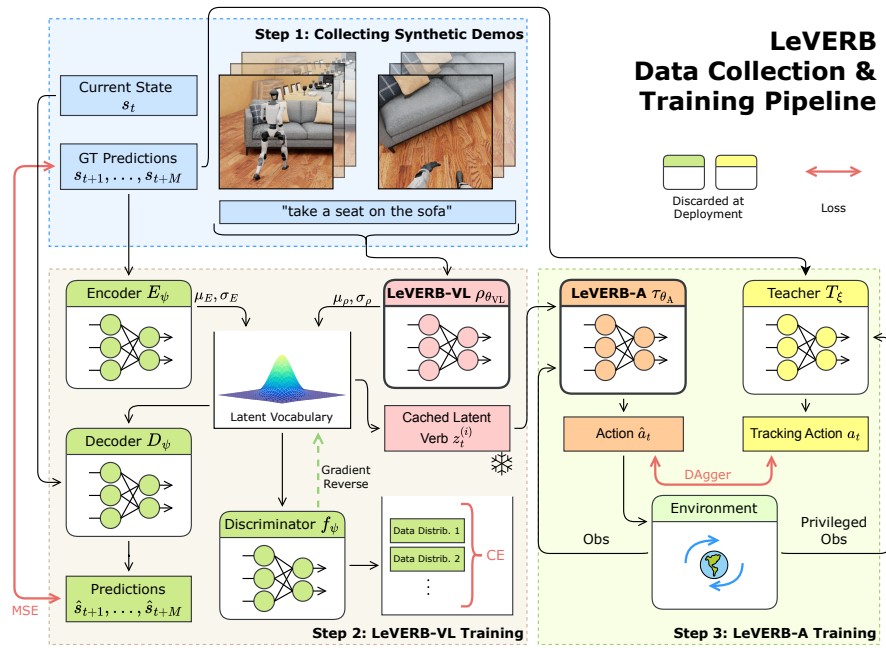

Figure 3: Details of our data collection and training pipeline. Step 1: we collect a synthetic, photorealistic dataset of retargeted motions in IsaacSim, and annotate with text instructions. Step 2: we train LeVERB-VL with a kinematic trajectory reconstruction task, and obtain a regularized latent verb vocabulary, from which we cache the latent verbs $z_t^{(i)}$ for every rollout $i$ in the dataset. Step 3: we use $z_t^{(i)}$ to condition LeVERB-A. It is DAgger-distilled from teacher tracking policy $T_\xi$, which receives future reference command $s_{t+1}$ that corresponds to the latent verb's intention.

with language instructions from a small number of kinematics motion trajectories, we employ a procedural generation pipeline to scale and randomize each rollout. Specifically, we randomize scene backgrounds, object properties, task setups, camera views, and mirror rollouts to ensure diverse and semantically rich data. We then label them with egocentric text commands manually or with a VLM (Lin et al., 2024a). The detailed workflow is introduced in Appendix A.

With 154 trajectories, each randomized 100 times, We generate 17.1 hours of photorealistic motion rollouts. Table 1 summarizes the mixture of different tasks in them. Each demonstration consists of images $I_0, \ldots, I_N$, a text instruction $c$, and robot kinematic states $s_0, \ldots, s_N$.

To further boost data diversity, we use a VLM to annotate text-only motion pairs without having to run photorealistic rendering. In total, we augment the vision-language data with 2.7 hours of language-only data covering 500 diverse trajectories. To address the lack of visual input, we inject spatial cues into the text (e.g., "the red chair on the left") to retain disambiguating context.

## 4 DUAL PROCESS HUMANOID CONTROL

As introduced in Figure 1, LeVERB follows a dual-process inference pipeline. In this section, we detail the design and training of our dual-process model architecture, depicted in Figure 3. We begin by outlining the hierarchical structure in Section 4.1, and then elaborate on the components of our system: LeVERB-VL (*System 2*) in Section 4.2, and LeVERB-A (*System 1*) in Section 4.3.

### 4.1 OVERALL MODEL HIERARCHY: DECOUPLING VISION-LANGUAGE AND ACTIONS

We formulate the VLA-driven WBC policy as $\pi_\theta(a_t \mid o_t)$, where $a_t$ is the dynamics-level action, $o_t = \left[o_t^{\mathrm{prop}}, I_t, a_{t-1}, c\right]^T$ is the observation, with $o_t^{\mathrm{prop}}$ is the proprioceptive sensor readings, $I_t$ is the visual inputs from a egocentric and a randomized third-person camera, and $c$ is the textual instruction.

Formally, our hierarchical system-1-system-2 policy is formulated at inference-time as

$$\pi_\theta(a_t \mid o_t) = \int \tau_{\theta_A}(a_t \mid z_t, o_t^{\text{prop}}, a_{t-1}) \cdot \rho_{\theta_{\text{VL}}}(z_t \mid I_t, c) \mathrm{d}z_t \tag{1}$$

where $\rho_{\theta_{\text{VL}}}$ denotes the high-level system 2 handling vision-language instruction and closed-loop visual feedback, which we name LeVERB-VL, and $\tau_{\theta_A}$ denotes the low-level system-1 action policy, which we call LeVERB-A. $\theta_{\text{VL}}$ and $\theta_A$ are policy parameters that corresponds to the two models.

A latent vector $z$ serves as a one-way interface from LeVERB-VL to LeVERB-A, and is at the core of LeVERB-VL training. Intuitively, the latent space of $z$ is a descriptive vocabulary that encodes complex whole-body motion objectives, and $z$ is a latent sampled from this vocabulary.

LeVERB-VL runs at $10\,\text{Hz}$, while LeVERB-A outputs joint position actions[1] at $50\,\text{Hz}$. This decoupling of vision-language and dynamics-level action information enables separated training of the two systems, avoiding the heavy computations for graphical rendering required in end-to-end approaches.

## 4.2 LeVERB-VL Training: Vision-Language-Action Semantic Alignment

The goal of LeVERB-VL is to map vision and language inputs into a smooth, regularized latent vocabulary space for motion control. To achieve this, we use a residual CVAE where the latent of a VLA prior with only vision-language inputs is combined with a privileged trajectory encoder to form a residual latent space. This encourages the VLA to focus on semantic reasoning while offloading motion-specific details to the trajectory encoder. The combined latent is then sampled to condition a decoder that predicts future poses $\hat{s}_{t+1}, \ldots, \hat{s}_{t+M}$ from the current state $s_t$. Finally, we introduce a discriminator that aligns data from different sources into a unified latent space.

**LeVERB-VL $\rho_{\theta_{\text{VL}}}$.** The VLA prior consists of three modules: a vision encoder, a text encoder, and a standard Transformer (Vaswani et al., 2017) backbone. For the vision encoder, we use the visual component of SigLiP (Zhai et al., 2023; Beyer et al., 2022), Since the vision and text encoders are contrastively pre-trained, the resulting embeddings are semantically aligned, which facilitates effective multimodal fusion. During training, images from two views—egocentric (head-mounted) and third-person—are independently processed by the frozen ViT-B/16 SigLiP visual encoder. The resulting image tokens are attention-pooled to produce image tokens $i_t^{\text{ego}}$ and $i_t^{\text{exo}}$, respectively. The vision encoder is pre-trained on WebLI (Chen et al., 2022) and remains frozen throughout training. The text encoder, also from the same SigLiP model, converts the textual instruction into a language token $l_t$. These tokens are concatenated to form the input sequence to the Transformer backbone: $\text{obs}_t = [l_t, i_t^{\text{ego}}, i_t^{\text{exo}}]$. Here, $t$ denotes the current time step; we only use observations from the current frame without any temporal history to reduce the risk of overfitting (Lin et al., 2024b; Mandlekar et al., 2021). The sequence $\text{obs}_t$ is then passed through the Transformer and a subsequent MLP head to predict the distribution over the observation, parameterized by the mean $\mu_\rho$ and variance $\sigma_\rho$. We ablate the size of the backbone Transformer model in Appendix B

**Kinematics Encoder $E_\psi$.** Since LeVERB-VL only observes the current vision-language inputs, we introduce a kinematics encoder to capture additional information from future states. The encoder is an MLP that takes the flattened ground-truth future states $s_{t+1}, \ldots, s_{t+M}$ as input and predicts the mean $\mu_E$ and variance $\rho_E$ of the latent distribution.

**Residual Latent Space**. Inspired by motion generation (Ling et al., 2020), we construct the latent distribution as $q(z_t \mid s_{t+1:t+M}, I_t, c, o_t)$, where the mean is a residual connection of the action encoder and the LeVERB-VL $\rho_{\theta_{VL}}$: $\mu = \mu_\rho + \mu_E$. The variance is taken directly from the action encoder: $\sigma = \sigma_E$. This setup allows the encoder to provide additional fine-grained information to help reconstruction, allowing the VLA to focus more on semantics. A KL loss is applied to this posterior to ensure the encoder captures only information not already inferable from vision-language inputs. During training, we apply the standard reparameterization trick to sample $z_t = \mu + \sigma \cdot \epsilon$, where $\epsilon \sim \mathcal{N}(0, I)$.

**Kinematics Decoder $D_\psi$.** A sampled latent $z_t$ from the posterior distribution and current state $s_t$ are fed into this MLP to reconstruct the future states $\hat{s}_{t+1}, \ldots, \hat{s}_{t+M}$.

---

[1]Joint position actions are not equivalent to the desired joint positions on humanoids, especially with contact dynamics and realizable stiffness and damping. They are re-parameterized torque-level actions.

**Discriminator** $f_\psi$. To leverage both vision-language and language-only trajectories from LeVERB-Bench, we mix these two sources during System 2 training by feeding a white-noise image to LeVERB-VL for the "blind" trajectories. However, this introduces a distributional shift between the latent embeddings of "blind" and "non-blind" inputs, which can hinder generalization. To align the latent spaces and encourage a shared representation, we introduce a discriminator inspired by O'Connell et al. (2022), which takes the latent $z_t$ as input and predicts whether an actual image was present. We apply a Gradient Reversal Layer (GRL) (Ganin & Lempitsky, 2015) during training, such that the adversarial learning encourages LeVERB-VL to produce modality-invariant latents, regardless of image availability.

**Training Objective**. Our final training objective consists of three components: trajectory reconstruction, distribution alignment, and adversarial classification. The full objective is

$$\mathcal{L}(\theta_{\text{VL}}, \psi) = \underbrace{\mathbb{E}_{\rho_{\theta_{\text{VL}}}(z|I,c,s_t)} \left[ \left\| D_\psi(s_t, z) - s_{t+1:t+H}^R \right\|_2^2 \right]}_{\text{trajectory reconstruction}} + \underbrace{\beta_1 \, D_{\text{KL}} \left( q(z_t) \, \| \, p(z_t) \right)}_{\text{distribution alignment}} + \underbrace{\beta_2 \, \mathcal{L}_{\text{disc}}(z_t)}_{\text{adversarial classification}} \,.$$

(2)

Here, $q(z_t) = \mathcal{N}(\mu_\rho + \mu_E, \sigma_E^2)$ is the action-conditioned posterior, and $p(z_t) = \mathcal{N}(\mu_\rho, \sigma_\rho^2)$ is the observation-conditioned prior from LeVERB-VL. We include data mixture strategy, data processing pipeline, hyperparameter selection and training recipe in Appendix B.

### 4.3 LeVERB-A Training: Distilling Actions with Learned Latent Distribution

After training LeVERB-VL, we freeze its latent space and train LeVERB-A by first learning vision-language-agnostic teachers, and then distilling a transformer-based student policy conditioned on the latent distribution from LeVERB-VL.

**Training Teachers** $T_\xi$. First, we train vision-language-agnostic teacher policies capable of accurately tracking different categories of retargeted kinematics trajectories from LeVERB-Bench. The policy receives privileged proprioceptive observations $o_t^{\text{priv}}$ and reference motions as commmands, and outputs expert actions $a_t$. We train teachers with Proximal Policy Optimization (PPO) (Schulman et al., 2017) and apply domain randomization for zero-shot sim-to-real transfer and early terminations to help training. Full details of teacher policies are provided in Appendix D.

**LeVERB-A** $\tau_{\theta_A}$. Next, we distill high-quality actions from multiple teacher policies into a unified student policy conditioned on latent commands generated by LeVERB-VL.

At the start of each episode, we sample a motion trajectory from LeVERB-VL's training set and extract the latent distribution's mean and variance, $\mu_{\rho,\text{traj}}$ and $\sigma_{\rho,\text{traj}}$. A random timestep $t$ is selected as the episode's start. Every $H$ steps, matching the System 1-2 resampling interval, we sample a latent code $z_t \sim \mathcal{N}(\mu_{\rho,t}, \sigma_{\rho,t})$ from the predicted distribution and hold it fixed until the next resampling. Importantly, we sample from the latent distribution rather than using the mean $\mu_\rho$. As LeVERB-VL captures a multimodal mapping between vision-language semantics and motions, using only the mean would impose a unimodal approximation, degrading policy performance.

LeVERB-A uses a Transformer that receives the observation $o_t^{\text{prop}}$ and latent code $z_t$ as separate tokens. It is trained via DAgger (Ross et al., 2011) with Huber loss against the teacher's actions $a_t$, which has better robustness to outliers. At deployment, LeVERB-A is conditioned on the predicted mean $\mu_\rho$ from LeVERB-VL. Additional details are provided in Appendix E.

## 5 EXPERIMENT

In this section, we evaluate LeVERB on multi-modal tasks with LeVERB-Bench. Since there exists no prior work that establishes a fair comparison with our method, we mainly present multiple ablated variants of LeVERB to demonstrate the effectiveness of our proposed method, including a naive dual-process VLA. Then, we showcase zero-shot real-world results on a humanoid robot hardware.

### 5.1 CLOSED-LOOP EVALUATION ON LeVERB-BENCH

First, we present closed-loop evaluations of LeVERB on various whole-body tasks from LeVERB-Bench. This setup closely matches real-world deployment and is used as the primary quantitative

Table 2: **Results of LeVERB against its ablated versions.** For each task/environment, we conduct 20 runs and report the success rate in percentages. Definitions of the abbreviations on both axes (e.g., ND, NE, NVL, VNF, VNR, VNS) are provided in Section 5.1.

| Tasks | Environment | LeVERB | ND | NE | NVL | NLS | NS | EI |
|---|---|---|---|---|---|---|---|---|
| *Vision-Language Tasks* | | | | | | | | |
| VNF | Objective | 80 | 75 | 75 | 15 | 0 | 0 | 10 |
| VNR | | 30 | 10 | 45 | 10 | 5 | 0 | 0 |
| VNF | Distractor | 75 | 55 | 60 | 0 | 0 | 0 | 10 |
| VNR | | 30 | 10 | 25 | 15 | 10 | 0 | 0 |
| VNF | Cluttered | 50 | 5 | 25 | 15 | 5 | 0 | 0 |
| VNR | | 25 | 0 | 5 | 5 | 5 | 0 | 0 |
| VNS | - | 5 | 0 | 5 | 0 | 0 | 0 | 0 |
| *Language-Only Tasks* | | | | | | | | |
| Sit | - | 100 | 0 | 100 | 40 | 5 | 10 | 5 |
| Stand | - | 90 | 75 | 90 | 55 | 10 | 15 | 0 |
| Locomotion | - | 100 | 100 | 100 | 100 | 25 | 50 | 70 |
| **All** | **-** | **58.5** | 33.0 | 53.0 | 25.5 | 6.5 | 7.5 | 9.5 |

results. We note that while individual items and textures are individually within the training distribution, their combinations are *unseen* from the training dataset.

**Ablation Variants**. We ablate the following components of the proposed architecture in Section 4:

- **No Discriminator (ND):** Removes the adversarial discriminator during LeVERB training.
- **No Kinematics Encoder (NE):** Removes the kinematics encoder $E_\psi$ from LeVERB-VL training, shifting the burden of encoding motion style entirely to the vision-language model.
- **No LeVERB-VL (NVL):** Directly conditions the low-level controller on visual and language embeddings, bypassing the high-level policy, similar to Shao et al. (2025).
- **No Low-level Sampling (NLS):** Uses mean instead of sampling in LeVERB-A training.
- **No Sampling (NS):** Disables sampling in both LeVERB-VL and LeVERB-A, yielding a naive hierarchical VLA baseline with deterministic conditioning on VL outputs.
- **Explicit Interface (EI):** Instead of feeding the latent verb to LeVERB-A, we pass it through decoder $D_\psi$ to obtain the decoded kinematic trajectory, and then train a separate whole-body controller student policy that takes local frame trajectory as tracking target. This is a naive system-1-2 approach with trajectory as an interface.

**Task Subcategories**. We sub-categorize each task in LeVERB-Bench to identify difficulty levels:

- **Visual Navigation – Front / Rear (VNF / VNR):** The navigation target is placed in front of the robot at spawn (easy), or behind it (hard), requiring a turnaround motion.
- **Objective / Distractor / Cluttered:** The scene includes only the navigation target, 1–2 distractor objects, or a fully cluttered environment.
- **Visual Navigation – Sit (VNS):** Includes walk to a target chair, turn around, and sit down.

Table 2 shows the success rate of LeVERB compared with its ablated versions. We evaluate each task on 20 scenes with unseen material and object combinations. We find that LeVERB stays on top in 9 out of the 10 task categories, reaching an average success rate of 58.9%. The NE variant shows slightly decreased performance at 53%, likely because its latent space is more fine-grained and contains less semantic information, which is more vulnerable to unseen scenes.

In contrast, the ND variant shows a significant decrease in performance in visual navigation tasks. This is likely due to the separation of the latent vocabulary distribution between the two categories of demos, vision-language ones and language-only ones, as a result of removing the discriminator designed to align data from different sources. This makes the model unable to apply the motion skills

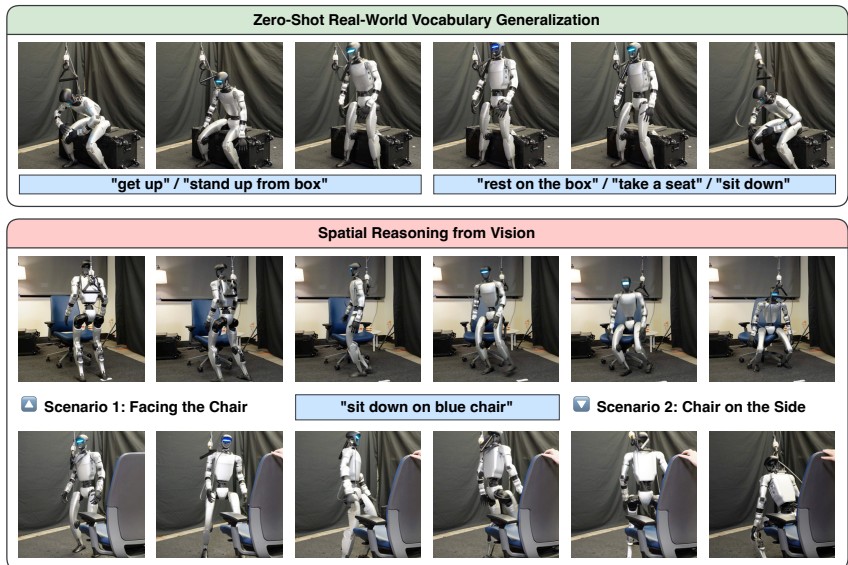

Figure 4: Top: LeVERB responds robustly to human vocabulary variations. Bottom: LeVERB executes different sit-down maneuvers conditioned on the chair's visual location, demonstrating spatial reasoning capabilities.

learned from language-only demos on vision-language tasks, thus resulting in smaller data coverage and less generalization. Similarly, we believe that the NS variant fails completely on almost all tasks because with a simple auto-encoder setup, its learned latent space is even more unstructured, leading to bad interpolation behavior and frequent OODs for the action module.

Furthermore, the NVL baseline achieves minimal success rates across all visual-language tasks. This shows that including the VLA planning capacity does help dramatically on visual-language input. In contrast, the performance on text-only tasks is much better, even 100% in locomotion. This aligns with prior work (Shao et al., 2025) that atomic language commands can be learned without high-level reasoning capacity. Lastly, we confirm that by not including the latent sampling in training LeVERB-A (NLS), the latent distribution cannot be fully captured by the low-level policy. Since this low-level policy is trained separately from LeVERB-VL, a mismatch in the interface distribution is likely to be detrimental.

Lastly, we demonstrate the necessity of a latent interface for whole-body VLA through the EI variant, which let a low level controller, trained the same way as LeVERB-A but instead conditioned on a future trajectory prediction, to track LeVERB-VL's explicit predictions. The categorical failure of this approach mostly comes from the low-frequency output of the high-level module, causing jittering and running out of control reference for the low-level controller, and resulting in constant catastrophic falling of the robot. In addition, the robot exhibits un-natural whole-body behavior, such as failure to keep a symmetrical walking gait and raising arms indecisively back and forth. This shows that explicit-interface-based method let the high-level vision-language module overwhelmingly dictates the low-level behavior, without ever experienced the dynamic-level intricacies, which lies in the expertise of the low-level whole-body controller. Coupled with the low frequency issue, this ablation variant highlights the necessity for a latent interface in humanoid whole-body control where the high-level module should provide **context** rather than direct control command.

## 5.2 REAL-WORLD DEPLOYMENT

Finally, we demonstrate the dynamics-level zero-shot sim-to-real transfer of LeVERB onto a real-world humanoid robot, Unitree G1. For visual navigation and locomotion task, the model can successfully close the vision feedback loop in the real world. Figure 4 demonstrates the proposed method in successfully replaying the visual navigation and sitting task in real world. In simulation, we give language commands with unseen combinations of verbs and objects (e.g. rest on the box), and record the closed-loop latent verbs outputted by LeVERB-VL, which demonstrates vocabulary

generalization abilities to correctly execute the correct motion. We also test the spatial reasoning ability of our method from vision by placing a target chair in different poses with respect to the robot's initial position, and have the robot walk and turn the appropriate amount to land in the chair by relying on visual feedback. We then open-loop replay the latent verbs in real world to LeVERB-A, executing the task successfully. This shows the dynamics-level sim-to-real readiness of LeVERB, enabling future real-world vision-in-the-loop deployment.

## 6 CONCLUSION

In this work, we present LeVERB, the first vision-language latent action model for humanoid whole-body control, and the first sim-to-real-ready, photorealistic benchmark for its kind. With a carefully designed dual-process, CVAE-based VLA, LeVERB can be zero-shot deployed to real, although trained only on a small synthetic dataset. In terms of task success rate, our method can outperform a naive hierarchical VLA by 7.8 times. For future work, we conjecture that a post-training pipeline, especially RL fine-tuning, could potentially improve policy performance by further aligning the closed-loop latent vocabulary distribution.

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

# A  DETAILED VISUAL-LANGUAGE WBC DATA GENERATION WORKFLOW

1. Create scene-level randomization: we select the color and material randomization options that can be applied to the objects in the background scene.

2. Create object-level randomization: we label daily household objects such as chairs and desks to randomize their color and material characteristics. This level of granularity is helpful for creating semantically meaningful task instructions such as "walk towards the yellow desk" or "go sit on the red sofa".

3. Create a task: given a prerecorded motion trajectory in the scene, we strategically place objective and distractor objects around the trajectory, such that a semantically meaningful task instruction can be given. For example, we spawn an objective in front of the end of the trajectory, such that this task can be labeled with "walk towards xx". The actual object placed and its properties are subject to randomization.

4. Procedurally generate variants: we let the simulator then randomize all visual features and task-related objects, and collect 100 demos for each trajectory. Each rollout features the onboard first-person-view camera, as well as 2-3 fixed third-person cameras, randomly positioned such that the entire task is within the camera frame.

5. Augment by mirroring: We mirror half the demos to boost data diversity.

# B  IMPLEMENTATION DETAILS FOR SYSTEM 2

**Data Mixture Strategy**  The training of System 2 follows a data mixture strategy as described in Table 3, comprising 3,696 trajectories with images and 2,300 trajectories without images. To enhance the robustness of the learned latent representation across diverse language styles and visual environments, we augment the dataset by repeating trajectories with varied language prompts and alternative image renderings. This results in a more balanced and diverse dataset across data sources and visual domains. Such augmentation helps mitigate overfitting to the limited trajectories used in System 2 training and improves the generalization of the latent space for downstream tasks.

Table 3: **Statistics of the data mixture recipe.**

| Category | Count | % | Unique Traj | Description |
|---|---|---|---|---|
| Total Demos | 5,996 | 100% | 614 | - |
| Vision Language | 3,696 | 61.6% | 154 | - |
| Language only | 2,300 | 38.4% | 460 | - |
| **Environment (Demonstrations with Images)** | | | | |
| Brown Stone | 456 | 12.3% | 19 | Apartment building with kitchens and living rooms |
| Living Room | 408 | 11.0% | 12 | Small living room |
| Modern House | 1,872 | 50.6% | 89 | Large house with kitchen, living room and bedroom |
| Kitchens | 960 | 26% | 34 | Small kitchens with partial enclosure for easy camera mounting |
| **Source (Demonstrations without Images)** | | | | |
| Whole Body (AMASS) | 425 | 18.5% | 85 | Reaching and sitting trajectories |
| Walk (LAFAN) | 520 | 22.6% | 104 | Egocentric and navigation trajectories |
| Run (LAFAN) | 1,115 | 48.5% | 219 | Running trajectories |
| RL Motions (In-House) | 240 | 10.4% | 52 | Egocentric trajectories |

**Data Processing**   For System 2 training, the pipeline processes proprioception and action for a 13-joint robotic system (1 root joint and 12 body joints) The proprioception data captures the state of each joint, where the root joint's pose is represented by its position $(x, y, z)$ in world coordinates and rotation (yaw, roll, pitch) in Euler angles, while the remaining body joints are represented by their $(x, y, z)$ positions only (w.r.t the root joint). The rotation information for the root joint is converted to a 6D rotation representation (Zhou et al., 2019) to ensure continuous and differentiable learning. The action space is designed to predict delta actions (changes) between future steps and current step, where the root joint's action includes both delta position and delta rotation, while other body joints only require delta position predictions. This design choice reflects the navigation task's requirements, where the root joint's full pose control is crucial for navigation, while other body joints primarily need position control. The current state $s_t$ is a vector including pitch, roll of the root joint and $x, y, z$ positions of the body joints. The future states $s_{t+1}, ..., s_{t+M}$ is the delta action mentioned above. To enhance robustness, we optionally apply noise to both proprioception and action data during training.

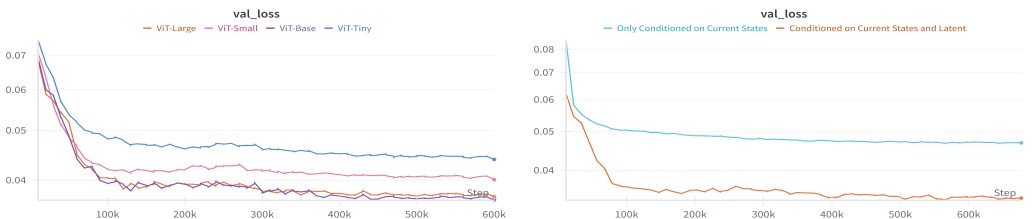

Figure 5: **The validation loss curves for training of system 2 of LeVERB** The left part shows the validation loss curve in Equation 2 with different size of backbone Transformer. The right part shows the validation loss of trajectory reconstruction for conditioned on different input.

**Hyperparameter Selection**   We use frozen ViT-Base models for both the visual and textual encoders, each producing 768-dimensional features. For the Transformer backbone in LeVERB-VL, we ablate model sizes ranging from ViT-Tiny to ViT-Base and show the validation loss in Figure 5 left side, and find that ViT-Base provides a good trade-off between performance and computational efficiency. All other components, including the latent representation, kinematics encoder, and kinematics decoder, operate in a latent space of dimension 256.

**Training Details**   For the training objective in Equation 2, we set $\beta_1 = 10^{-1}$ and $\beta_2 = 5 \times 10^{-4}$. To stabilize training, we apply schedulers to both the distribution alignment and adversarial classification terms. Each scheduler acts as an additional scaling factor that linearly increases from 0 to 1 over the first 40% of training epochs. These hyperparameters are selected based on empirical performance observed during ablation studies. We use 2 NVIDIA Ada 6000 GPUs to train system 2 of LeVERB with the global batchsize of 512. The total trainable parameter of system 2 is 102.56 million parameters (ViT-Base LeVERB-VL backbone).

## C   EFFECTIVENESS OF THE CVAE OBJECTIVE

Since we provide the CVAE decoder with current states, it is a valid concern that the decoder might be able to reconstruct the immediate future states without the latent input, i.e. the reconstruction objective of immediate future states is not an effective objective to construct a meaningful latent space. We disprove this concern by running a variant with randomly sampled latents from a normal distribution and current state $s_t$ are fed into decoder $D_\psi$. As shown in right side of Figure 5, this variant converges to a much higher imitation loss, showing that the information excluding the latent input for the decoder is insufficient to reconstruct future states, thus our training objective is effective.

## D   DETAILS FOR LEVERB-A TEACHER POLICIES

In this section, we introduce the details for training the teacher imitation policies.

**Observations and Actions**  The observations for the teacher policy include two parts: proprioceptive observations and commands related to reference motions. For proprioceptive observations, we include base linear velocity, base angular velocity, and joint positions and velocities. The commands include reference joint positions and velocities in the next frame, and relative position and orientation of the reference torso link in the next frame with respect to the actual one. We also include previous actions as input. We define the actions in joint position with small stiffness and damping.

Table 4: Reward Terms and Formulations

| **Reward Terms** | | | |
| --- | --- | --- | --- |
| **Reward Name** | **Weight** | **Mathematical Formulation** | $\sigma$ **(if any)** |
| Global Torso Position | 0.5 | $\exp\left(-\frac{\|\mathbf{p}_{\text{motion}}-\mathbf{p}_{\text{robot}}\|^2}{\sigma^2}\right)$ | $\sqrt{0.25}$ |
| Global Torso Orientation | 0.3 | $\exp\left(-\frac{\text{quat\_error}(\mathbf{q}_{\text{motion}},\mathbf{q}_{\text{robot}})^2}{\sigma^2}\right)$ | $\sqrt{0.5}$ |
| Global Body Position | 0.5 | $\exp\left(-\frac{1}{\sigma^2}\|\mathbf{x}_{\text{motion}}-\mathbf{x}_{\text{robot}}\|^2\right)$ | $\sqrt{0.25}$ |
| Joint Position Error | $-1$ | $-\|\boldsymbol{\theta}_{\text{motion}}-\boldsymbol{\theta}_{\text{robot}}\|$ | $-$ |
| Joint Velocity Error | $-0.1$ | $-\|\dot{\boldsymbol{\theta}}_{\text{motion}}-\dot{\boldsymbol{\theta}}_{\text{robot}}\|$ | $-$ |
| Action Rate L2 | $-0.001$ | $-\|\mathbf{a}_t-\mathbf{a}_{t-1}\|^2$ | $-$ |
| Joint Limit Violation | $-100.0$ | $-\mathbb{I}_{\text{violate\_limit}}$ | $-$ |
| Termination Signal | $-200.0$ | $-\mathbb{I}_{\text{done}}$ | $-$ |
| **Termination Terms** | | | |
| **Name** | **Type** | **Mathematical Formulation** | **Parameter** |
| Bad Reference Position | Termination | $\|\mathbf{p}_{\text{motion}}-\mathbf{p}_{\text{robot}}\|>\tau_{\text{pos}}$ | $\tau_{\text{pos}}=0.5$ |
| Bad Reference Orientation | Termination | $|\text{proj}_z(\mathbf{g}^B_{\text{motion}}-\mathbf{g}^B_{\text{robot}})|>\tau_{\text{ori}}$ | $\tau_{\text{ori}}=0.8$ |
| **Domain Randomization** | | | |
| **Name** | **Mode** | **Range Type** | **Range** |
| Ground Property | Startup | Friction | $[0.3,0.8]$ |
| Ground Property | Startup | Restitution | $[0,0.5]$ |
| Joint Default Pos | Startup | Position Offset | $[-0.05,0.05]$ |
| Joint Armature | Startup | Armature Scale | $[0.2,2.0]$ |
| Push Robot | Interval: $[10,15]$s | X-Y Velocity | $[-0.5,0.5]$ |

**Rewards and Early Termination**  We formulate the reward function with three parts. First, we include DeepMimic (Peng et al., 2018)-style motion tracking rewards. Second, we include smoothing terms including action rate penalties and soft joint limits set to 90% of the hard joint limits. Last, we penalize the policy when it terminates due to large tracking error. Specifically, the episode is terminated when the tracking error of either the position or the orientation of the torso link is too large. We summarize these reward functions in Table 4.

**Domain Randomization**  For zero-shot sim-to-real transfer, we randomize physics properties including ground friction and restitution, joint default positions in calibration and joint armature in training the teacher policies. We also include a velocity perturbation term. These terms are summarized in Table 4.

**Architecture**  For each teacher, we use a 3-layer MLP with hidden dimensions of 512, 256, and 128. ELU is used as the activation function, except for the final layer, which has no activation.

## E  DETAILS FOR LEVERB-A STUDENT POLICIES

In this section, we introduce the details for learning the student policy with DAgger.

**Observations**   Similar to the teacher policies, for proprioceptive observations, we include base linear velocity, base angular velocity, previous actions, and joint positions and velocities from the joint encoders. In addition, we also include a gravity vector projected onto the base link as an observation of the row and pitch angles of the robot. For command, we include sampled latent from the output of the System 2 VLA. Since System 2 runs at 10Hz and System 1 runs at 50Hz, for every 5 steps, we sample a new latent vector from the latent Gaussian distribution of that timestep, and keep it fixed for the next 5 steps. For actions, we use the as the teachers.

**Early Termination and Domain Randomization**   In order to keep the teacher policy within its training distribution so that the expert actions are optimal, we include the same early termination conditions and domain randomizations for the student policy. These terms are summarized in the last two blocks in Table 4.

**Architecture**   For the student, we use a Transformer with 2 layers, 4 attention heads, and a hidden dimension of 128. The model encodes the latent command and proprioceptive inputs as separate tokens, with a dropout rate of 0.3 applied to the attention weights. ELU is used as the activation function throughout. Notably, in this setting, we find that incorporating observation history degrades performance, as the policy could infer future actions from prior observations alone, reducing the effectiveness of the latent commands.

## F   DETAILS FOR DEPLOYMENTS

**Hardware Platform**   We deploy the policy on a standard G1 robot from Unitree. We use the official SDK to obtain the sensor reading and send the action as the desired position on the joint command while setting the desired velocity to zero, kp, and kd to the stiffness and damping in the simulation.

**LeVERB-A**   System 1 getting sensor information from the joint encoder, IMU with Unitree SDK, and custom state estimator at $500\,\text{Hz}$; the inference happens in onboard CPU of the robot at $50\,\text{Hz}$ with ONNX runtime, all the code is implemented in C++ to fulfill the real-time performance requirements. The latent command interface is exposed as a ROS2 topic. The onboard RealSense camera image is also compressed and streamed with a ROS2 topic so that any device in the network has access.

**LeVERB-VL**   System 2 runs on an external desktop PC with NVIDIA RTX 4090 GPU at about $10\,\text{Hz}$, its input is from a third-person-view camera connected by USB, and the onboard camera on the robot, both of which are running at 30 FPS and with the resolution of $1080 \times 720$ pixels. A input text prompt window is shown on the PC screen. The policy outputs the latent verb, which is broadcasted on a ROS2 topic.

## G   EVALUATION ENVIRONMENT

We evaluate the tasks on 20 random environments and instructions for each task category. The texture and object properties of the scene are completely randomized and previously unseen. The third-person camera angle is locally randomized to the degree allowable by the scene. We ensure that every task in the evaluation are visually unseen in the training dataset.

