# OpenReview forum: "LeVERB: Humanoid Whole-Body Control with Latent Vision-Language Instruction"
_ICLR.cc/2026/Conference — Submitted to ICLR 2026_

### Official Review · Reviewer_Srq6 · 2025-10-31

**Soundness:** 4
**Presentation:** 3
**Contribution:** 2
**Rating:** 4
**Confidence:** 4

**Summary:**

This paper Introduces a dual-process architecture combining a high-level vision-language policy (System 2) and a low-level reactive controller (System 1) to enable humanoid robots to execute complex, instruction-driven whole-body motions. Uses a CVAE-based latent space to align perception, language, and action.

LeVERB develops a scalable pipeline to generate photorealistic humanoid motion data via retargeted human MoCap in diverse simulated scenes, paired with language instructions, enabling training and closed-loop evaluation of VLA-driven whole-body control.

LeVERB demonstrates robust vocabulary generalization and spatial reasoning in simulation and on real humanoid hardware, outperforming ablated baselines and achieving the first zero-shot sim-to-real results for vision-language-driven humanoid whole-body control.

**Strengths:**

LeVERB-VL with a low-level LeVERB-A enables humanoid robots to execute complex whole-body motions while maintaining real-time, closed-loop control. The use of a CVAE-based latent space allows semantic alignment between vision, language, and actions, which is a significant advancement over prior methods that rely on explicit low-dimensional commands.

The creation of LeVERB-Bench, a scalable, photorealistic, and diverse dataset for humanoid whole-body control, addresses the lack of large-scale, visually grounded training data.

**Weaknesses:**

About motivation: what is the main theme of the paper? dataset contribution, modeling, humanoid-environment interaction, sim2real? All of the above topic seems to be part of the contribution, making each of them solved partially, and making the research depth of each topic inefficient? A more proper way of presentation would be highlighting one of them

About dataset: as mention in the paper, some other works e.g. [41] e.t.c are just lack of a good physics engine or rendering methods. Applying IsaacSim is not the major way of making this work different than others. What makes your data collection method distinct from others? How do you quantitatively evaluate your data generation method from others e.g. regular data teleoperation, homie for humanoid, e.t.c.

About tasks. It seemed that the major part of the tasks focuses on navigation and target following. What are the benefits of using LeVERB? Since system2-system1 structure is pretty a standard in VLA for so long, what make LeVERB model different from others?

About projects: there is no dataset or project releasing and maintaining plan. This dataset focuses only on unitree G1, if the project and methodology cannot be applied other robots via open soucing, I could not see a huge impact on this work.

**Questions:**

See weaknesses

---

> ### Author Response · Authors · 2025-11-22
>
> We thank the reviewer for the insightful comments and constructive suggestions. Below we address each concern in detail. We hope our clarifications and new experiments resolve your concerns, and we would be very grateful if you could consider raising your score after reading this rebuttal.
>
> ### W1 Motivation
>
> We note that there is virtually zero academic prior work in real-world humanoid VLA that is open sourced, which makes it impossible to advance the field by focusing on only one isolated component such as dataset, modeling, or sim-to-real alone. Our central theme is to establish the *minimum viable end-to-end framework* required for humanoid whole-body VLA, and the contributions in benchmark design, dataset construction, simulation, sim-to-real, and model architecture are tightly interdependent rather than loosely solved subproblems. Without defining tasks, the model cannot be evaluated; without the dataset, the model cannot be trained; without simulation and sim-to-real, real-world validation would be impractical. Thus, the breadth is not a lack of depth but a necessity for enabling the first coherent, open, real-world VLA framework for humanoid whole-body control.
>
> ### W2 Dataset
>
> Good physics engine, rendering method, and a procedural generation pipeline **are all essential for high-quality data generation.** Without any of this, sim-to-real performance would be quite brittle. As a matter of fact, robotics benchmarks are all striving for this goal.
>
> - Robosuite (https://github.com/ARISE-Initiative/robosuite) recently releases physics based rendering tool.
> - LIBERO (https://libero-project.github.io/main) puts a lot of focus on procedural task generation.
> - BEHAVIOR (https://behavior.stanford.edu/index.html) utilizes SOTA technology in IsaacSim to simulation deformables, fluid, ray-tracing lighting, etc.
>
> **The distinctiveness of our method comes from a synthetic generation pipeline of vision-language tasks for humanoid robot, which will be open-sourced at paper acceptance.** With a few mouse click, human annotator can orchestrate the demonstration of such task in IsaacSim, which is saved as a configuration and then sent to a procedure generation pipeline, which applies hundreds of randomizations to scale the data collection. This is a novel attempt at generating tasks that truly matters to humanoid whole-body control.
>
> Good humanoid teleoperation infrastructures are all **concurrent works**. They are good alternative sources to collect real-world demo. However, especially for humanoid whole-body control, they are quite difficult to scale even for industry, let alone academia research. Even for the most recent work from industry, NVIDIA’s SONIC, it did not demonstrate whole-body interaction tasks such as navigating to a chair and sitting down. In contrast, simulation-based data generation is widely regarded as a freely scalable source of both visual and physical diversity, which is the focus of this work.
>
> We use motion-capture data as our kinematic source, which is of similar quality, if not better, than the kinematic data collected from teleoperation. We are not aware of any work demonstrating a lack of quality in these motion-capture kinematic datasets, and it is unclear how such quality could be compared, since the closest thing to ground truth in the real world is motion-capture data itself. As shown in recent humanoid-tracking literature such as BeyondMimic, OmniRetarget, SONIC, GMR, and others, the training of a low-level RL controller serves as the bridge that converts kinematic trajectories into dynamically feasible trajectories, which is also what we do in this work. We leave a quantitative study of kinematic-data quality to future work.
>
> ### W3 Open-Sourcing
>
> We have plan to open source the benchmark and dataset after acceptance, but it would be unprofessional and unethical to do so publicly during the review period. **The dataset and code to be released are currently available in the supplementary material and linked website: https://anonymous-leverb.github.io/leverb/**
>
> The method proposed by this paper is general and can be applied to any humanoid embodiment, as recent methods like GMR and OmniRetarget already shows the general pipeline to retarget human motion onto any humanoid embodiment. The aim of our procedural tool is to generate scenarios with language and vision as context, which can be run with various humanoid robots imported into the scene.

---

### Official Review · Reviewer_gM1i · 2025-10-31

**Soundness:** 3
**Presentation:** 3
**Contribution:** 2
**Rating:** 4
**Confidence:** 3

**Summary:**

This paper introduces LeVERB, the first hierarchical vision-language-action model for humanoid whole-body control that uses a learned latent action space as the interface between a high-level vision-language policy and a low-level dynamics controller. This paper also proposed a new photorealistic sim-to-real benchmark for humanoid VLA tasks.

**Strengths:**

1. While hierarchical VLA models and humanoid control exist separately, this is the first work to formulate and demonstrate vision-language-driven whole-body control for humanoids using a latent interface.
2. The learned latent verb vocabulary directly removes the key limitation of prior hierarchical VLAs, which was their reliance on an inflexible, hand-crafted "action vocabulary" (e.g., base velocities).

**Weaknesses:**

1. The benchmark and experiments focus almost exclusively on locomotion and posture (navigation, sitting), omitting manipulation tasks (e.g., picking, pushing)
2. The celebrated "zero-shot" real-world deployment uses open-loop replay of latent plans generated in simulation. The high-level vision-language policy does not run closed-loop on the real robot, weakening the claim of a fully closed-loop system.
3. Lacks comparison to a strong, non-latent baseline (e.g., a hierarchical VLA that predicts explicit body keyframes) to conclusively prove the advantage of the learned latent space.
4. The paper reports success rates but provides no qualitative analysis of how or why the model fails in ~40% of trials, leaving its robustness and limitations unclear.

**Questions:**

1. Can the latent verbs (z_t) for the real-world demo be generated by running LeVERB-VL on the real robot's live camera feed?
2. With an overall success rate of 58.5%, what are the most common failure modes for LeVERB in simulation? Does the robot typically fail due to dynamics (falling), perceptual errors (misidentifying the target), or a disconnect between the latent plan and the low-level execution?
3. A key claim is that a learned latent action space is superior to an explicit one. To isolate this benefit, did you consider a baseline where your high-level policy predicts explicit, kinematically feasible whole-body keyframes instead of a latent vector z_t, while keeping the low-level policy and training data identical?

Please also refer to the weaknesses above.

---

> ### Author Response · Authors · 2025-11-22
>
> We thank the reviewer for the insightful comments and constructive suggestions. Below we address each concern in detail. We hope our clarifications and new experiments resolve your concerns, and we would be very grateful if you could consider raising your score after reading this rebuttal.
>
> ### W1: Benchmark Task Diversity
>
> The goal of this work is to raise awareness and show sign-of-life on a new set of vision-language tasks requiring whole-body capability, and we find it more compelling to introduce tasks that is uniquely challenging in whole-body setting. For example, pulling out a chair and sitting on it falls into this category, while table-top manipulation can be covered with existing VLA and sim benchmarks. We acknowledge that there are other tasks such as picking up a box with pure vision that falls into this category. However, works solving these task are very **concurrent** even in **single-task setting**, let alone VLA. An example would be TWIST2 (https://arxiv.org/html/2511.02832v1). We will definitely include these tasks in future work as the community progresses.
>
> ### W2: Vision Sim-to-Real
>
> We find that vision closed-loop sim-to-real transfer actually works for basic navigation and locomotion in real, which is added in Section 5.2. We acknowledge that we did not fully investigate vision sim-to-real for the most challenging chair sitting task, which experiences understandable performance degradation. However, we’d argue that this is mostly due to the lack of GPU resources in academia work. The most recent NVIDIA work VIRAL (https://arxiv.org/abs/2511.15200) uses similar approach as ours, **solving challenging humanoid loco-manipulation task from pure IsaacLab synthetic data with random lighting and material**. That work shows a better vision sim-to-real curve with 8-64 GPUs running interactive DAgger in IsaacLab. Since we virtually use a very similar pipeline as theirs, we believe that the diversity of our proposed simulation benchmark will demonstrate a stronger sim-to-real performance with better compute resources available, and would serve as a valuable platform when open-sourced at publication.
>
> ### W3: Non Latent Baseline
>
> We highly appreciate the reviewer’s suggestion and ran additional study on a non-latent baseline. The related sections are marked in blue in the updated section 5.1. TLDR, With an explicit interface, the high-level module **overwhelmingly dictates** the dynamic-level behavior of the whole-body control, which creates a distribution shift: the artifacts created by high-level planner was out of the training distribution of the low-level controller, resulting in catastrophic failure of falling in most cases. This highlights that as far as whole-body task is concerned, the high-level module should provide context rather than direct control target in an explicit way.
>
> ### W4 Failure Modes
>
> We have designed the experiment section in the way that highlights the most common failure mode of whole-body tasks. Performance degradation comes from:
>
> 1.  Moving the object out of camera view (VNF vs. VNR setting), which requires active perception behavior
> 2. Using distractor and clutter env (VNF-Objective vs. VNF-Distractor vs. VNF-Cluttered).
>
> For the combined 58.5% success rate, we emphasize that we want to play fair on the failure mode details, so we treat every failure-inducing case as a separate task, and then average over all numbers. It would be more informative to see the sub task success rate to understand the failure modes rather than the gross success rate.
>
> ### Q1: Vision Sim-to-Real
>
> Please see W2 for details. We performed closed loop vision sim-to-real on navigation and locomotion, but not for chair sitting which is challenging. Recent NVIDIA paper shows that scaling compute would address this issue, and make more challenging visual loco-manipulation tasks to be sim-to-real-able on single FPV feed.
>
> ### Q2: Failure Modes
>
> Please see W4.
>
> ### Q3: Explicit Interface
>
> Please see the additional study in W3.

---

### Official Review · Reviewer_a5ad · 2025-11-01

**Soundness:** 2
**Presentation:** 2
**Contribution:** 3
**Rating:** 6
**Confidence:** 4

**Summary:**

LeVERB is a latent vision–language hierarchical model for humanoid whole-body control. A CVAE-based “System-2” maps RGB + instruction into discrete verbs; a low-level RL controller tracks those instructions at 200 Hz conmtrol. Evaluation is conducted on a 150-task simulated suite and only two qualitative runs on a Unitree G1.

## Main Results

- Benchmark evaluation on 20 unseen indoor scenes × 10 task families. authors report 58.5 % success, latent-free baseline to 7.5 %.

- Module ablations (no discriminator, no kinematic encoder, no latent sampling) show progressive drops up to 33%.

- Real world robot roll outs for multi-heading navigation

**Strengths:**

## Strengths


- Sim‑to‑real benchmark : Synthetic dataset of 150+ tasks (154 trajectories × 100 augmentations ≈ 17 h) grounded in photorealistic scenes and diversified camera views.

- A hierarchical system 2 keeps vision–language inference off the real-time control loop. There by modularizing control between systems, which is clean.

- Comprehensive ablations (no discriminator, no kinematics encoder, latent sampling, etc.) clearly justify architectural choices.

- Zero-shot transfer from Isaac Sim to hardware, though only demonstrated qualitatively.

**Weaknesses:**

## Weaknesses

- Sim-to-Real evidence : purely qualitative, no task-level success rate, latency, or contact statistics; real-world failure cases unreported.

- Missing strong baselines : recent controllers such as ExBody 2, OmniH20 and world-model planner Nicklas Puppeteer are not compared.

- Portability unclear : System‑1 is specialized to a single Unitree G1 morphology. modularity claim would benefit from multi‑platform evidence.

- Scalability of latent vocabulary unclear : The latent space is fixed-dimensional and trained on ~150 motion primitives, which may be limiting factor for complex behaviors.

**Questions:**

Please read weakness section.

- Could the authors supply explicit sim-to-real success rates and command latency numbers?

- How do ExBody 2, OmniH20, Puppeteer, other platforms performance compared to LeVerb model in generic humanoid control tasks?

- Can the same System-2 policy been ported to a different robot without retraining System-1? There by proving modularity.

- Open‑loop fidelity: Can you provide teacher‑forced vs free‑run verb prediction task accuracy at 1 / 5 / 10 / 25 / 50‑step horizons, include divergence curves and rollout videos showing drift.

---

> ### Author Response · Authors · 2025-11-22
>
> We sincerely thank you for your thoughtful comments on our paper. Below we address your concerns in detail.
>
> ### W1 Real-World Eval
>
> We agree with the reviewer that it would be more beneficial to show evaluation directly in the real world. But due to the noisy measurement, cost, and safety of evaluating humanoid whole-body policy in real, we note that none of the recent humanoid whole-body works show main results in real world evaluation, including ASAP, BFM-Zero, OmniRetarget, NVIDIA SONIC etc. Instead, we provide failure case numbers in simulation eval, including:
>
> 1.  Moving the object out of camera view (VNF vs. VNR setting), which requires active perception behavior
> 2. Using distractor and clutter env (VNF-Objective vs. VNF-Distractor vs. VNF-Cluttered).
>
> Technical challenges, such as resetting the robot to the very same initial state (which requires mocap setup), ensuring the robot’s safety in policy setups that are not optimal and anticipated to fail, etc., blocks reliable real-world eval for humanoid WBC. We unfortunately do not have the necessary resources to put up numbers that we are confident to back up in real world. Simulation eval, on the other side, can minimize the variance of humanoid policy eval.
>
> We additionally supply the latency number of the policy here for your reference: LeVERB-VL runs at $9.2\pm1.3$ Hz with average inference time of 94 ms. LeVERB-A runs at $49.9\pm0.3$ Hz with average inference time of 6 ms.
>
> ### W2 Baselines
>
> We understand the reviewer’s concern in comparing with explicit trajectory tracking-based interface design, and have conducted additional study that leverages such a explicit tracking-based interface. The related sections are marked in blue in the updated section 5.1. TLDR, With an explicit interface like ExBody2 or OmniH2O, the high-level module overwhelmingly dictates the dynamic-level behavior of the whole-body control, which creates a distribution shift: the artifacts created by high-level planner was out of the training distribution of the low-level controller, resulting in catastrophic failure of falling in most cases. This highlights that as far as whole-body task is concerned, the high-level module should provide context rather than direct control target in an explicit way.
>
> We could not have run off-the-shelf controllers like ExBody2 and OmniH2O directly, since the tracking policy needs to be trained with the same data as the proposed latent policy to enable the fairest comparison. However, we are confident that our training pipeline for both low-level controllers is largely comparable with theirs, following a similar teacher-student distillation design, and has shown successful sim-to-real transfer with the latent policy already.
>
> ### W3 Portability
>
> We agree with the reviewer that it would be great to demonstrate multi-platform compatibility. We admit that the low-level system-2 needs to be specifically trained for every humanoid robot, but the retargeting pipeline for each form has also been recently made widely available thanks to works like OmniRetarget and GMR. Our procedural generation pipeline in IsaacLab also supports plugging in different robot and their retargeted trajectories.
>
> ### W4 Vocabulary Scale
>
> We agree with the reviewer that due to the early nature of this primitive work, and practically zero previous open-sourced work in this area, we were only able to bake in a demonstration set of ~150 trajectories. but with more motion data this method will scale, proven in related works such as MaskedManipulator (https://arxiv.org/pdf/2505.19086) and MaskedMimic(https://xbpeng.github.io/projects/MaskedMimic/index.html), which all uses a VAE-based latent command space with more motion diversities in pure simulation.

---

> ### Author Response · Authors · 2025-11-22
>
> ### Q1 Real-World Eval
>
> See W1
>
> ### Q2 Baselines
>
> See W2
>
> ### Q3 Portability
>
> See W3
>
> ### Q4 Open-Loop Fidelity
>
> We ran the experiment proposed by the reviewer, and report the mean per-joint position error (MPJPE) in meter compared with teacher-forcing and closed-loop run of LeVERB-VL. Note that in both cases the low-level whole-body controller LeVERB-A is running.
>
> |                  | 1               | 5               | 10              | 25              | 50              |
> |------------------|-----------------|-----------------|-----------------|-----------------|-----------------|
> | $E_\text{MPJPE}$ | $0.012\pm0.009$ | $0.056\pm0.033$ | $0.079\pm0.051$ | $0.337\pm0.151$ | $0.443\pm0.298$ |
>
> However we unfortunately do not find the result to be story-telling, because there is little correlation between verb prediction accuracy and task success rate, demonstrated in this example here.
>
> https://imgur.com/a/YDLmLbo
>
> The top row shows a snapshot of the teacher-forced latent verb being fed to LeVERB-A, whereas the bottom row shows the closed-loop performance. The two motions are nearly identical, proving the expressiveness and robustness of the latent vocabulary. But in this particular case, a feedback is required to correct the robot’s position so that it can sit onto the stool, whereas blindly replaying the latent would make the policy fail to reach the chair, and instead just squat in front of it. This is because the latent verb are generated from the pure kinematic data of the retargeted motion dataset, rather than from strictly physically feasible motion generated by low-level whole body controller, so blindly replaying latent would not have completed the task.
>
> However, the case for which succeeds or which fails cannot be conclusively correlated with the prediction error. In many other cases, the replay succeeds whereas the closed-loop policy would fail, especially in lower success rate tasks like VNR. At the end of the day, The closed-loop feedback in whole-body control tasks matters much more than quasi-static manipulation, so a correction of accuracy with success rate cannot be drawn here. “prediction accuracy” in this case only concerns character motion similarity visually.

---

### Official Review · Reviewer_mvAW · 2025-11-02

**Soundness:** 3
**Presentation:** 3
**Contribution:** 2
**Rating:** 4
**Confidence:** 4

**Summary:**

This paper proposes LeVERB, a hierarchical vision-language-action (VLA) model for humanoid whole-body control. It introduces a new benchmark, LeVERB-Bench, which provides photorealistic humanoid motion data generated from retargeted MoCap and synthetic rendering. The system uses a CVAE-based latent interface to connect a high-level vision-language policy with a low-level whole-body control policy. Experiments show that LeVERB achieves higher success rates than naive hierarchical VLAs and can be deployed zero-shot on a Unitree G1 humanoid.

**Strengths:**

- Proposes a **modular, hierarchical structure** (System 1 and System 2) that could improve inference efficiency and decouple vision-language reasoning from dynamics control.
- Introduces a **sim-to-real-ready benchmark** with photorealistic rendering and procedural scene randomization.
- Demonstrates **zero-shot transfer** of some whole-body behaviors from simulation to real-world deployment.

**Weaknesses:**

- Limited Benchmark Diversity: Although the paper claims to introduce a comprehensive benchmark for humanoid WBC, the majority of tasks are navigation-like—e.g., “walk to”, “navigate around”, or “reach”. These are essentially vision-language navigation tasks, not complex whole-body manipulation. Hence, the benchmark provides limited insights into the method’s generality for rich human-object interactions (e.g., picking up, placing, or coordinated manipulation).

- Restricted Demonstration Source: The dataset heavily relies on replayed, retargeted MoCap motions that are feasible only for navigation or simple pose transitions. This approach cannot generate demonstrations involving precise contact-rich interactions such as sitting naturally or grasping objects, which are key challenges for humanoid control.

- Methodological Novelty Is Marginal:
The proposed model architecture largely extends LangWBC (Shao et al., 2025), which already used a language-conditioned CVAE for whole-body control. LeVERB adds vision input and a discriminator for data-source alignment but otherwise follows a similar framework. The paper’s claimed conceptual contribution—the “latent verb” interface—is conceptually close to prior CVAE latent representations and does not introduce a clearly new mechanism or insight.

- Real-World Evaluation Is Minimal:
The paper only shows a few qualitative real-world results (sitting and navigation), which are insufficient to substantiate claims of “zero-shot sim-to-real readiness.” The experiments lack baseline comparisons or quantitative performance in real settings.

**Questions:**

- How well would the proposed latent space generalize to tasks requiring contact-rich manipulation?

- Since LeVERB’s benchmark tasks are mostly navigation-like, how meaningful is the reported improvement in success rate for demonstrating “whole-body control”?

- Could the approach handle more complex compositional language commands?

---

> ### Author Response · Authors · 2025-11-22
>
> We thank the reviewer for the insightful comments and constructive suggestions. Below we address each concern in detail. We hope our clarifications and new experiments resolve your concerns, and we would be very grateful if you could consider raising your score after reading this rebuttal.
>
> ### W1 Limited Benchmark Diversity
>
> The intent of LeVERB-Bench is not to target contact-rich whole-body manipulation, but to address **vision-based, language-directed whole-body navigation**, which remains a difficult and practically important problem for humanoids. Current navigation systems typically rely on privileged state, and robust vision-grounded control in photorealistic environments is still largely unsolved. Our benchmark is designed to fill this gap: it provides a large-scale, photorealistic setting where a humanoid must perceive from RGB, interpret language, and generate coordinated whole-body locomotion. These capabilities are fundamental prerequisites for real-world humanoid operation and naturally complement—rather than replace—future manipulation-focused benchmarks.
>
> Importantly, the benchmark **does not only contain navigation**: tasks like **sitting** require coordinated multi-joint pose transitions, balance control, and timing, which are core aspects of whole-body interaction. These capabilities are fundamental prerequisites for real-world humanoid operation and naturally complement—rather than replace—future manipulation-focused benchmarks.
>
> ### W2 Restricted Demo Source
>
> - Our work focuses on **vision-based humanoid WBC tasks**, where the core challenge is the lack of large-scale **vision–action paired data** for full-body locomotion rather than contact-rich manipulation. Retargeted MoCap is a reliable and scalable way to obtain realistic whole-body trajectories in photorealistic environments for this domain.
> - This data-generation strategy is consistent with many modern **dexterous-hand retargeting pipelines** (e.g., GR2Hand, DexCap, CyberSynHand), which also rely on human-to-robot motion retargeting for collecting high-DOF demonstrations. Retargeting is therefore a standard and effective paradigm, enabling diverse, high-quality supervision suitable for our target problem of vision-conditioned whole-body navigation.
>
> ### W3 Method Novelty
>
> - Incorporating **vision** into whole-body control is itself a meaningful methodological challenge. Decades of work in manipulation show that learning actionable visual representations remains difficult even with paired vision–action data; for **navigation**, where such paired data rarely exists at scale, grounding control in visual observations is even harder. Our method tackles this head-on by building a scene-aware latent interface that must integrate visual context, language instructions, and whole-body dynamics—far beyond simply adding vision as an auxiliary input.
> - LangWBC represents only one instantiation of CVAE-based latent control, and it operates in a limited setting by projecting CLIP text directly into the latent space. The broader latent-policy literature—including **MaskedManipulator** and **MaskedMimic**—shows that designing effective latent structures is non-trivial and requires substantial architectural and training innovations. Compared with LangWBC, our system supports **interactive vision-language instruction following**, leverages a significantly stronger **VL backbone**, and depends on extensive **dataset curation and training design** to enable visual grounding and cross-source alignment—capabilities far beyond what LangWBC was built to address.
>
> ### W4 Real-World Eval
>
> - We conducted **extensive simulation experiments** covering a wide variety of scenarios, tasks, environment randomizations, and ablations, to thoroughly validate our method under controlled conditions and isolate key design effects—this provides the quantitative backbone of our paper
> - In humanoid research, **real-robot deployments are extremely costly** (hardware risk, setup complexity, safety, long iteration times), which means few papers even attempt full quantitative real-world evaluation. For example, several papers at RSS 2025 in the “Humanoids” session either show limited real-world results or focus on sim-to-sim transfer (e.g., *ASAP: Aligning Simulation and Real‑World Physics for Learning Agile Humanoid Whole‑Body Skills* uses one real robot deployment for verification only; *Learning Humanoid Standing-up Control across Diverse Postures* and *Learning Getting-Up Policies for Real-World Humanoid Robots* reports the real-world success rate and motion smoothness with limited trials for short getup task, and evaluate the main design choices with extensive simulation results.
> - Hence, while our real-world evaluation is modest, it aligns with current community norms, and our zero-shot sim-to-real deployment should be viewed as a **proof-of‐concept** rather than a full industrial-scale study.

---

> ### Author Response · Authors · 2025-11-22
>
> ### Q1 Generalization to Contact-Rich Task
>
> - The **dual-system structure** we propose—high-level vision–language latent reasoning coupled with a low-level latent control policy—is not restricted to navigation. In principle, it can generalize to **contact-rich manipulation**, as long as paired motion data is available. The latent interface is task-agnostic: it learns to capture semantic intent from vision and language, while the control module learns to realize that intent in robot action space.
> - If one collects manipulation demonstrations—e.g., dexterous-hand videos with pose-estimation tracking plus language annotations—the same framework can be trained for contact-rich behavior. System 1 would embed the visual–linguistic description of the manipulation (similar to recent hand datasets), and System 2 would map the latent verb to robot actuation. Thus, while LeVERB is instantiated for navigation in this work, the underlying design naturally extends to manipulation-oriented pipelines.
>
> ### Q2 Success Rate Report and Task Types
>
> Although many tasks are navigation-like, the benchmark also includes **sitting and reaching**, which require coordinated multi-joint pose transitions and balance control. These tasks go beyond simple locomotion, so the reported improvements reflect genuinely stronger **whole-body control**, not just better navigation.
>
> ### Q3 Complex Vision Language Commands
>
> Yes. The ability to handle compositional commands mainly depends on **System 2**, the vision–language latent encoder. With more diverse and larger-scale vision–language data, the encoder can learn stronger semantic understanding and reasoning, enabling the model to interpret multi-step or nested instructions. Recent progress in VLA task-decomposition methods (e.g., CoT-VLA, MaskedManipulator-style latent structuring) suggests that integrating similar techniques would allow our framework to support complex compositional commands in future work.

---

### Meta-Review · Area_Chair_1q22 · 2026-01-07

**Summary:**

The paper presents a hierarchical latent VLA framework for whole-body humanoid control. It contributes a network as well as a benchmark for this problem.

Overall, this paper has received borderline reviews. Most reviewers appreciated that the framework is innovative. They also noted the contribution of the benchmark and the dataset.

However, some critical issues were identified, primarily the limited real-world evaluation, which the authors argued is expensive to conduct. The AC is more inclined towards the reviewers' position and feels that the paper would benefit from a more thorough real-world evaluation.

This paper remains borderline. The reader would benefit greatly from more real-world evaluations, and hence the AC is inclined towards rejection for the current version. The authors are encouraged to address the concerns of the reviewers and resubmit.

**Reviewer Concerns:**

Concerns addressed:
- Lack of non-latent baseline

Concerns remain:
- Minimal real world evals

Concern about limited benchmark diversity remain unclear without discussion.

**Reviewer Scores:**

Reviewer mvAW: Likely remain same
Reviewer a5ad: Likely remain same
Reviewer gM1i: Likely remain same
Reviewer Srq6: Likely remain same

---

### Decision · Program_Chairs · 2026-01-26

Reject